# *Angiotensin-Converting Enzyme* Gene Polymorphisms and Diabetic Neuropathy: Insights from a Scoping Review and Scientometric Analysis

**DOI:** 10.3390/diseases13090289

**Published:** 2025-09-01

**Authors:** Rafaela Cirillo de Melo, Paula Rothbarth Silva, Nathalia Marçallo Peixoto Souza, Mateus Santana Lopes, Wellington Martins de Carvalho Ragassi, Luana Mota Ferreira, Fabiane Gomes de Moraes Rego, Marcel Henrique Marcondes Sari

**Affiliations:** 1Undergraduation in Pharmacy, Federal University of Paraná (UFPR), Curitiba 80210-170, PR, Brazil; rafaelacirillo@ufpr.br; 2Graduation Program in Pharmaceutical Sciences, Federal University of Paraná (UFPR), Curitiba 80210-170, PR, Brazil; p.rothbarth@hotmail.com (P.R.S.); nathaliamarcallo@gmail.com (N.M.P.S.); masanlo@outlook.com.br (M.S.L.); well.ace1@gmail.com (W.M.d.C.R.); luanamota@ufpr.br (L.M.F.); rego@ufpr.br (F.G.d.M.R.)

**Keywords:** single-nucleotide polymorphism, hyperglycemia, angiotensin-converting enzyme gene, diabetic complications, biomarkers

## Abstract

Background/Objectives: Diabetic neuropathy (DN) is one of the most common and disabling complications of diabetes mellitus (DM), affecting motor, sensory, and autonomic nerves. Genetic factors, particularly polymorphisms in the *Angiotensin-converting enzyme* (*ACE*) gene, have been proposed as contributors to DN susceptibility. This study aimed to synthesize the scientific evidence on *ACE* gene polymorphisms and their association with DN through a scoping review combined with scientometric analysis. Methods: A comprehensive search of PubMed, Scopus, and Web of Science was performed in February 2025, following JBI and PRISMA-ScR guidelines. Observational studies involving individuals with DN and the genotyping of ACE polymorphisms were included. Scientometric mapping was conducted using the Bibliometrix package in RStudio to identify publication trends and key thematic terms. Results: From 100 screened articles, 11 met the inclusion criteria. Most studies (72.7%) addressed diabetic peripheral neuropathy, while 27.3% investigated cardiac autonomic neuropathy. All studies analyzed the I/D polymorphism in intron 16 of the *ACE* gene. The D allele and DD genotype were associated with increased susceptibility to DN in over half of the studies (6/11), while the II genotype was reported as protective in 3/11. Findings varied by ethnicity and study design. The scientometric analysis identified ‘peripheral diabetic neuropathy’, type 2 diabetes’, and ‘ACE gene polymorphism’ as the most frequently co-occurring terms, indicating that research on this topic has been concentrated around these themes, while showing limited diversity in geographic origin and scope. Conclusions: *ACE* I/D polymorphism appears to modulate susceptibility to DN, though interethnic variability and methodological heterogeneity challenge definitive conclusions. Broader, standardized studies are needed to validate its utility as a predictive biomarker.

## 1. Introduction

In 2024, the International Diabetes Federation (IDF) reported that 589 million people worldwide live with diabetes [1], a condition encompassing a spectrum of metabolic disorders characterized by hyperglycemia resulting from either impaired insulin secretion, defective insulin action, or a combination of both mechanisms [2]. *Diabetes mellitus* (DM) is primarily classified into three types: type 1 *diabetes mellitus* (T1DM), accounting for approximately 5–10% of cases, is caused by the autoimmune destruction of pancreatic β-cells [2,3,4]; type 2 *diabetes mellitus* (T2DM) represents the majority of cases, around 90–95%, and is marked by insulin resistance and a relative deficiency in insulin secretion [2,3]; and gestational *diabetes mellitus* (GDM), meanwhile, refers to varying degrees of hyperglycemia that occur during pregnancy [1,2]. The classification of DM is closely linked to its etiology, which involves environmental factors such as obesity, physical inactivity, and smoking. Additionally, genetic predisposition and metabolic syndromes play significant roles in the disease development [5,6,7].

Chronic hyperglycemia is associated with long-term damage, dysfunction, and failure in different organs, especially the eyes, kidneys, nerves, heart, and blood vessels. In this sense, glycemic control management plays a crucial role in preventing the development of microvascular complications. Microvascular complications include nephropathy, retinopathy, and diabetic peripheral neuropathy (DPN). The development of these microvascular complications is a significant risk factor for macrovascular diseases, such as stroke, heart failure, and atherosclerosis [3,8,9]. Diabetic neuropathy (DN) is the most prevalent chronic complication in individuals with DM, impacting roughly 50% of this population [10,11]. This condition involves nerve damage characterized by both structural and functional impairments, leading to symptoms such as pain and numbness. DN affects various nerve fibers, including motor, sensory, and autonomic pathways, resulting in a multifaceted clinical presentation [12]. A complex interplay of factors, including age, sex, and ethnicity, influences the prevalence of DN. It is noteworthy that individuals with T2DM are more frequently affected by DN compared to those with T1DM and may present complications at the time of diagnosis or even during a prediabetic state. However, the sensitivity of the diagnostic methods used affects the reported prevalence [10]. In addition, the number of young individuals diagnosed with T2DM has been increasing, contributing to these findings [13,14].

Among the intricate and complex factors, genetic predisposition plays a crucial role in DN pathogenesis [8,9,12,15]. In recent years, research into gene polymorphisms has yielded significant insights into the molecular pathways associated with susceptibility to DN, thereby facilitating the development of personalized prevention and treatment strategies. Of particular importance, scientific literature has consistently shown that various genetic variants contribute to DN predisposition. Some studies covered multiple genes, such as *THEG5*, *XIRP2*, *CSMD1*, *PPARA*, *EDN1*, *MTHFR, GPx*-1, *CAT, GSTM1*, *GSTT1*, *IL-10*, and *NOS1*, and highlighted the association between DN and their single-nucleotide polymorphisms (SNPs) [12,16]. While these findings underscore multifactorial genetic architecture, the *angiotensin-converting enzyme* (*ACE*) gene variants retain clinical and mechanistic relevance [15].

The ACE is part of an important regulatory mechanism of the renin–angiotensin system (RAS), which catalyzes the conversion of angiotensin I (Ang I) to angiotensin II (Ang II), a potent vasoconstrictor agent [17,18]. High Ang II levels are associated with impaired glucose and insulin regulation, thereby increasing the risk of developing DM [6,19]. Furthermore, *ACE* gene directly interfaces with the renin–angiotensin system (RAS), a pathway implicated in microvascular dysfunction, oxidative stress, and neuroinflammation, which are processes in DN pathogenesis [17,18]. The *ACE* gene, spanning 21 kb, is located at 17q23 and comprises 28 exons and 25 introns (Figure 1). Among the potential genetic variants of this gene, the insertion/deletion (I/D) polymorphism of *ACE* gene, also known as rs1799752 [20,21], is one of the most studied and is characterized by the presence (insertion, allele I) or absence (deletion, allele D) of a 287 bp Alu repeat sequence in intron 16, producing three genotypes (II homozygote, ID heterozygote, and DD homozygote) [18,22].

It was reported that individuals homozygous for the deletion/deletion (DD) allele have higher tissue and plasma ACE concentrations than ID heterozygotes and II homozygotes, which determine the level of ACE in plasma and tissues. Previous studies have shown that *ACE* (I/D) polymorphism, particularly the *ACE*-DD genotype, was associated with increased levels of plasma ACE and increased vasoconstriction, which would result in increased risk of T2DM [6,17,22]. In this context, polymorphisms are well known to influence gene expression and determine phenotypic variability, which can result in increased susceptibility to various diseases or a differential response to environmental factors [6,8,23]. Accordingly, some studies investigate the association between gene polymorphism and peripheral neuropathy [24], which could be a risk factor for cardiac autonomic neuropathy (CAN) in patients with diabetes [19]. Given the multifactorial nature of DM and its accompanying vascular complications [25], elucidating the genetic factors that influence disease susceptibility and progression is pivotal for enhancing personalized treatment paradigms. Notably, the insertion/deletion (I/D) polymorphism in the *ACE* gene has been identified as a potential genetic marker, with numerous studies indicating its role in the pathogenesis of DN [6,8,25]. Nevertheless, the results are inconsistent across different populations, likely arising from genetic diversity, methodological differences, and clinical heterogeneity. In addition, we highlight that this point has been underscored in a recent systematic review, which shows that heterogeneity in DN genetics often stems from differences in definitions of DN, genotyping methods, and the limited representation of diverse populations [16].

In this framework, systematically mapping the existing evidence is crucial for identifying knowledge gaps and guiding future investigations. Therefore, the major goal of this manuscript was to comprehensively compile and analyze the association between *ACE* gene polymorphisms with the susceptibility, progression, and severity of DN through a scoping review and scientometric evaluation. The review synthesizes information on genotyping methodologies, allele frequency distributions, and associated clinical findings, thereby enhancing our understanding of genetic predisposition and its implications for managing diabetes-related complications. 

## 2. Materials and Methods

### 2.1. Methodology

The scoping review was conducted in accordance with the guidelines set by the Joanna Briggs Institute, and the results are presented according to the Preferred Reporting Items for Systematic Reviews and Meta-Analyses Extension for Scoping Reviews (PRISMA-ScR) Checklist (Appendix A) [26,27]. The study protocol has been registered on the Open Science Framework (OSF) to ensure thorough documentation, and a PRISMA-ScR checklist is included as Appendix A. Further details are available at https://doi.org/10.17605/OSF.IO/J9KHN (24 January 2025).

### 2.2. Search Strategy

A comprehensive literature search was performed in the PubMed, Scopus and Web of Science databases (Table 1) in February 2025. Search terms related to *ACE* gene, polymorphisms and DN were combined using the Boolean operators “OR” and “AND”. To improve the thoroughness of the literature review, manual supplementary searches were performed to identify and evaluate pertinent studies that may not have been captured in indexed databases (list of references).

### 2.3. Eligibility Criteria

Utilizing a Population–Concept–Context (PCC) framework, this review examines individuals with DM (both type 1 and type 2) and diabetic cohorts where neuropathic outcomes have been evaluated (Population). It explores the genetic variation in the ACE gene and its association with susceptibility to DN, as well as its impact on the progression and severity of both peripheral and autonomic manifestations (Concept). This review is situated within the context of human observational research (including case–control or cohort studies) that incorporates ACE genotyping and assesses neuropathic endpoints (Context). The eligibility criteria were established based on a conceptual framework focusing on the *ACE* gene polymorphisms and their potential link to DN in individuals with DM. The primary aim was to investigate the relationship between genetic variations in the *ACE* gene, specifically the I/D polymorphism, and susceptibility to DN, encompassing both peripheral and autonomic manifestations. The key research question driving this investigation was “*How do ACE gene polymorphisms influence the susceptibility, progression, and severity of diabetic neuropathy?*”. This review incorporated studies that evaluated the prevalence of *ACE* polymorphisms within diabetic cohorts and their correlation with neuropathic outcomes.

The studies included in this review met the following criteria: (1) observational designs, such as case–control or cohort studies, involving individuals with DN (peripheral, autonomic, or other subtypes); and (2) identification of the genotypic profile and polymorphisms in the *ACE* gene. Studies were excluded if they were based on preclinical data (in vitro or preclinical studies), were unavailable in full text, or were categorized as editorials, letters to the editor, literature reviews, conference abstracts, or publications in non-Roman scripts (e.g., Russian, Korean, Japanese). 

### 2.4. Study Selection

Initially, a comprehensive search was conducted across the previously specified databases, and the results were imported into the Rayyan (Rayyan Systems, Inc., Doha, Qatar, free version) [28]. Duplicate records were identified and removed. In the first screening stage, titles and abstracts were blindly reviewed by two researchers to assess eligibility according to the predefined inclusion and exclusion criteria. Articles considered potentially relevant proceeded to the second stage, in which full-text versions were evaluated for final inclusion. Any disagreements between the reviewers were resolved through discussion or consultation with a third reviewer. Studies that did not meet the eligibility criteria were excluded, and the reasons for exclusion were systematically documented (Appendix A). Figure 2 depicts the overall process of identification, screening, eligibility assessment, and inclusion. 

### 2.5. Data Extraction

The selected articles underwent a full-text review, during which the following data were extracted and tabulated: authors, year of publication, study design, ethnicity, research objective, type of DN, genotyping methodology, sample size, *ACE* gene polymorphism, and main findings (Table 2). To enhance clarity and facilitate interpretation, key interrelationships identified across the studies were visually represented in graphical formats (Figure 3). A narrative synthesis was conducted based on descriptive statistics to elucidate the major contributions of each study, support comparative analyses among them, and investigate potential explanations for any observed discrepancies or gaps. Quantitative normalization of clinical or laboratory data was not conducted, as this review aimed to synthesize existing findings qualitatively. Allele and genotype associations were extracted as reported by the original studies. No harmonization was performed across different genetic models (dominant, recessive, and additive). The variability observed across studies was addressed by evaluating methodological variations and the reported associations, rather than through statistical normalization. Then, as this review did not involve a quantitative synthesis, formal sensitivity analyses were not applicable. Instead, we assessed robustness by qualitatively comparing patterns across diverse populations, study designs, and diagnostic criteria. This descriptive approach enabled us to identify areas of convergence in our findings, as well as instances where heterogeneity limited the strength of our conclusions.

### 2.6. Scientometric Analysis

A scientometric analysis was conducted to complement the synthesis of evidence by providing an overview of the scientific landscape regarding *ACE* gene polymorphisms and their association with DN [38,39]. The selected articles were compiled into a bibliographic database and exported to Microsoft Excel and analyzed using the Bibliometrix package (version 4.0.0) in R software (version 4.2.0) within RStudio (version 2024.06.14). The analysis focused on identifying the most frequently cited terms in titles, abstracts, and keywords, such as “neuropathy”, “*ACE* gene polymorphism” and “T2DM” (Figure 4), as well as visualizing the hierarchical structure and co-occurrence of these terms (Figure 5). 

## 3. Results

The search identified 100 articles, of which 16 were from the PubMed database, 47 from Scopus, and 37 from Web of Science. After submitting the studies to the Rayyan platform, 39 duplicates were removed, leaving 61 articles selected for preliminary screening. Upon analyzing the titles and abstracts, 38 articles were excluded because they did not meet the eligibility criteria. From the initial search results, a total of 23 articles were selected for an in-depth full-text review. Of these, 14 studies were excluded based on specific criteria, each exclusion being thoroughly documented (Appendix A). Consequently, nine studies fulfilled the established inclusion criteria. Furthermore, a manual search yielded two additional articles that also met these criteria. A total of 11 studies were included for critical evaluation in this review. Key data from these studies were systematically extracted and are presented in Table 2.

Figure 3 provides descriptive statistics for the included studies, presenting the countries of origin, year of publication, genotyping methodology, and DN type. In terms of geographic distribution, the studies were conducted in diverse populations, including those from Japan, Turkey, Spain, Pakistan, India, Iraq, Australia, Egypt, and England, reflecting a broad international interest but also highlighting regional gaps in the literature (Figure 3A). The studies included in this review were published between 2002 and 2025, indicating a sustained yet sporadic scientific interest in the association between *ACE* gene polymorphisms and DN over the past two decades (Figure 3B). However, there were notable gaps in the literature during specific periods, with few publications addressing the topic. Regarding the type of DN, the most frequently analyzed condition was DPN (8/11), followed by CAN (2/11) (Figure 3C,D). The genotyping methodologies employed were predominantly based on PCR techniques, with variations including conventional PCR, Taq-PCR, PCR-RFLP, and PCR-MADGE (Figure 3E). Lastly, the (I/D) polymorphism in intron 16 of the *ACE* gene was the primary genetic variant investigated in all included studies. No other polymorphism was examined, as the research consistently concentrated on the I/D variant, particularly the association of the D allele and the DD genotype with increased susceptibility to DN, and the II genotype with a potentially protective role (Figure 3D,F).

The scientometric analysis involves a comprehensive evaluation of the most cited terms and the hierarchical referencing of data within the literature. This assessment not only elucidates key themes in the domain but also underscores the critical need for intensified research on genetic polymorphisms associated with diabetic complications. The term “Neuropathy” is central to this review, as it represents the principal complication discussed across all the studies featured (Figure 4). The predominant terms identified pertain to the gene encoding the relevant enzyme. The descriptors “diabetic” and “type” are frequently encountered due to the prevalence of diabetic complications and their association with specific diabetes classifications. Additionally, “peripheral” and “polymorphism” are secondary terms that are referenced intermittently throughout the literature.

Figure 5 illustrates the proportional representation and hierarchical significance of data through the dimensions and areas of geometric shapes. The size of the rectangles serves as a quantitative indicator of the relevance of specific terms or expressions within the context. The most prominent terms identified include “peripheral diabetic neuropathy” and “type 2 *diabetes mellitus*,” which hold the highest significance. These are followed in relevance by “ACE *gene* polymorphism”, “angiotensin-converting enzyme”, “diabetic”, “genetic polymorphism” and “type 2 diabetes.” The prominence and selection of these terms are indicative of the thematic focus commonly encountered in relevant literature.

## 4. Discussion

This review included studies that investigated the potential association of *ACE* gene polymorphisms as risk/protective biomarkers for the development of DN. Accordingly, polymorphisms in this gene, particularly the I/D variant in intron 16, have been widely associated with alterations in serum and tissue ACE levels [20]. Patients with the DD genotype exhibit significantly higher ACE activity than those with the ID or II genotypes [32,34,35]. The increase in ACE activity amplifies the conversion of Ang I into Ang II, a potent vasoconstrictor and pro-inflammatory peptide with multifaceted implications in the pathophysiology of DM and its vascular complications. Elevated Ang II levels contribute to endothelial dysfunction, inflammation, and oxidative stress, primarily by activating NADPH oxidase, a major source of reactive oxygen species (ROS) in vascular tissues [20]. In DM patients, which is characterized by chronic hyperglycemia, impaired nitric oxide bioavailability, and microvascular vulnerability, there is a significant increase in oxidative stress that exacerbates the vulnerability of microvascular structures, leading to progressive injury of the *vasa nervorum*, the microvessels that are crucial for maintaining perfusion to peripheral nerves [33]. Additionally, Ang II receptor overactivation intensifies peripheral vasoconstriction, further compromising microcirculatory flow and peripheral nerve health [20].

Altogether, this molecular network suggests a plausible mechanistic link between *ACE* gene polymorphisms and DN development. The D allele, particularly in the DD genotype, is associated with increased ACE activity, resulting in elevated Ang II levels. This increase in angiotensin II is significant as it not only causes vasoconstriction and diminishes endoneurial blood flow but also heightens oxidative stress via the activation of NADPH oxidase and mitochondrial pathways. Additionally, Ang II activates pro-inflammatory signaling through the nuclear factor kappa B (NF-κB) pathway and enhances cytokine release, thereby contributing to fibrosis and extracellular matrix remodeling via the activation of transforming growth factor-β (TGF-β). Collectively, these mechanisms compromise neural microvascular supply, exacerbate axonal injury, and amplify the deleterious effects of chronic hyperglycemia. In contrast, the I allele, which is linked to lower ACE activity and correspondingly reduced Ang II levels, may offer some protective effects by dampening these maladaptive pathways. This biological framework provides a coherent explanation for the clinical associations observed between ACE genotypes and the risk of DN. 

Remarkably, the I/D polymorphism not only influences susceptibility to DN but may also modulate its severity and progression. Thus, the *ACE* gene represents a key intersection point between genetic predisposition, metabolic dysfunction, and microvascular complications in DM patients, underscoring its relevance as both a biomarker of risk and a potential therapeutic target [6,17,18,20]. In this context, a study by Ramathan et al. (2024) investigated patients with T2DM, conducting a comparative analysis between individuals with DPN and those without this complication [20]. The findings revealed a higher prevalence of the D allele among T2DM patients with DPN compared to the control group, suggesting a potential association between the D allele and an increased risk of developing DPN. Furthermore, the DD genotype frequency was significantly elevated in both T2DM patients with and without DPN when compared to controls. Individuals with the DD genotype exhibited higher serum ACE levels and activity. In these patients, the conversion of Ang I to II was enhanced, and the Ang II subsequently activates NADPH oxidase, the primary source of reactive oxygen species in vascular tissue, thereby contributing to the DN pathogenesis. However, the II genotype may be associated with a protective factor against DPN, as it alters the metabolic functions of nerve tissue, which could help prevent nerve damage. Additionally, the study showed that the II genotype was more prevalent in the control group than in patients with DPN. In addition, there was no significant difference in the frequencies of I and D alleles between the study groups. Accordingly, other studies have also shown that patients with the D/D genotype have a significantly increased risk of developing DPN compared to those with the I/D and I/I genotypes, suggesting that the D/D genotype is a genetic biomarker for predisposing individuals to DPN [32,35]. 

In contrast, a different genetic profile was found in the Turkish population [30,34]. The I/I genotype was associated with the development of hypertension in patients with DPN [32]. The study also identified the D/D genotype as a factor of susceptibility to DPN. Additionally, a statistically significant difference in the I/I genotype was observed in hypertensive patients with DPN compared to those without hypertension. The authors suggested that the D allele of the I/D polymorphism in the *ACE* gene may be associated with an increased risk of developing DPN. At the same time, the higher frequency of the I/I genotype among hypertensive patients with DPN may indicate a predisposition to hypertension in these individuals. The I/D polymorphism in the *ACE* gene may be associated with susceptibility to DPN and hypertension in diabetic patients, highlighting the importance of considering genetic factors in the assessment. However, another study obtained a different result by evaluating the Turkish population, patients with DN, and hypertension. Degirmenci and colleagues (2005) [39] found no significant difference in genotypes and allele frequencies of the I/D polymorphism in the *ACE* gene between T2DM patients and healthy controls. Although the difference in ACE activity between hypertensive and normotensive diabetic patients was not significant, higher ACE activity was found in hypertensive patients. Furthermore, the D allele was more frequent in patients living with T2DM.

Regarding DN patients, the I/D genotype prevalence was higher, D/D was intermediate, and I/I was lower compared to the control groups. Therefore, the authors suggest that I/D is a risk factor, while I/I is a protective factor against DN. Meanwhile, a study conducted in the Japanese population found an association between the polymorphism and the I allele in patients with polyneuropathy, suggesting a potential risk associated with the allele frequency and the development of polyneuropathy. The D allele, on the other hand, demonstrates a protective effect against this condition. However, the genotype distribution did not exhibit significant differences between patients with and without polyneuropathy [29].

A study conducted on the Pakistani population with T2DM and DPN did not identify a significant difference in the genotypic distribution of the I/D polymorphism in the *ACE* gene but suggested that the I/I genotype may be a protective factor for the development of DPN in T2DM patients [33]. However, the number of participants with DPN was substantially lower than that of those without the complication (276 vs. 476), resulting in an unbalanced genotypic distribution that may have impacted on the reliability and statistical power of the comparisons. 

Other studies investigated the association of polymorphisms in the *ACE* gene associated with CAN in populations from Iraq [19] and Australia [36]. In the Australian population, no significant association was observed between the ACE genotype and CAN [36]. On the other hand, the DD genotype and D allele were significantly more frequent in T2DM patients compared to healthy individuals in the Iraqi population [17]. Patients with CAN exhibited a higher frequency of the D allele compared to those without CAN, suggesting that the D/D genotype and the D allele in the I/D polymorphism of the *ACE* gene may indicate susceptibility to the development of T2DM. Additionally, the limited exploration of CAN suggests an underrepresentation of this clinically significant neuropathy form, despite its established impact on cardiovascular morbidity and mortality in DM patients. The lack of association found in studies focusing on CAN may reflect distinct pathophysiological mechanisms or a need for more sensitive diagnostic tools in genetic research involving autonomic dysfunction. 

Collectively, the analysis of these studies suggests a link between the DD genotype and an increased risk of developing DN, indicating that the D allele may act as a susceptibility factor [19,20,31,32,34,35]. In contrast, the II genotype has been identified in several investigations as a potential protective factor against this condition [20,30,33]. However, some studies reported no statistically significant associations between these genotypes and neuropathy, which could be attributed to methodological discrepancies, imbalanced sample sizes, or ethnic heterogeneity among the populations analyzed [36,40,41].

The recurrent examination of the *ACE* I/D gene polymorphism underscores its possible significance as a genetic marker concerning DN. Moreover, clinical observations have indicated that comorbidities, such as hypertension, can influence the phenotypic expression of specific genotypes [30,34]. This is particularly evident in patients with II genotypes who also exhibit hypertensive neuropathy [34]. The exploration of CAN has been limited, underscoring the need for more targeted and methodologically rigorous research in this subtype. Therefore, as illustrated in Figure 6, the relationship between ACE I/D genotypes and DN exhibits notable variability across different populations. For instance, studies involving cohorts from Japan, Iraq, Egypt, and Pakistan have identified an increased susceptibility linked to the DD genotype, which aligns with higher levels of ACE activity and angiotensin II that may exacerbate microvascular injury. In contrast, research conducted among Turkish and Spanish cohorts suggested a protective effect associated with the II genotype. At the same time, studies involving British, Indian, and Australian populations found no statistically significant association. This inter-population variability underscores that the contribution of the ACE polymorphism to DN risk is not uniform but rather influenced by a combination of various factors. These findings highlight the intricate interplay between genetic polymorphisms and the clinical manifestations of neuropathy, underscoring the critical need for future studies that incorporate clinical covariates and stratification based on relevant population subgroups.

Additionally, some studies have highlighted sex-related differences, showing an increased risk associated with the D allele in female subjects [36]. In this sense, a study conducted with English Caucasians with T2DM demonstrated that women carrying the D allele had a significantly increased risk of developing DPN, compared to those homozygous for the I allele [31]. In contrast, no significant association was observed in men, suggesting that sex-specific factors, such as hormonal differences, particularly the modulatory effect of estrogen on *ACE* gene expression, may mediate the impact of the D allele on DN risk. 

The scientometric analysis identified a significant thematic convergence in the literature, highlighted by recurring terms such as “peripheral diabetic neuropathy,” “type 2 diabetes,” “angiotensin-converting enzyme (ACE),” and “gene polymorphism.” These trends indicate ongoing scientific engagement with the genetic underpinnings of diabetic complications, especially those associated with microvascular damage [7,38,39]. Notably, despite the biological rationale supporting the role of *ACE* gene variations in DN, the relatively sparse number of publications adhering to rigorous methodological parameters suggests that this field remains a niche and underexplored area. The limited variety of key terms and research clusters, as shown by hierarchical mapping, further emphasizes the focus on a few genetic variants, predominantly the I/D polymorphism. In contrast, other pertinent *ACE* polymorphisms and interactive pathways have been largely overlooked in the context of DN.

Additionally, the analysis reveals significant geographic and temporal disparities in the literature. While research has emerged from nine countries, there is a notable concentration in specific regions, particularly Turkey and South Asia, alongside a conspicuous underrepresentation of populations from Africa and Latin America. This uneven distribution may exacerbate heterogeneity in findings, considering the role of ethnicity in genotype prevalence and expression. Furthermore, the irregular publication trajectory over the past two decades, characterized by intermittent output, suggests that research on *ACE* polymorphisms in DN has struggled to sustain momentum, potentially due to issues with reproducibility or the rise of alternative genetic targets. Collectively, these insights underscore the need for research that is geographically inclusive, thematically diverse, and methodologically robust to enhance our understanding of the genetic risk factors associated with DN.

The results of this review indicate significant heterogeneity in the genetic polymorphism profiles among populations [42,43], likely stemming from variations in methodology, genetic diversity [41,44], ethnic backgrounds, and environmental factors [5,8]. Ethnicity was identified as a critical factor influencing genotype distribution, which hampers the establishment of a consistent and generalizable association framework. This underscores the need for population-specific genetic mapping to enhance risk stratification and inform the development of targeted preventive strategies for DN. Future research should prioritize enhanced standardization in phenotype classification and ensure the inclusion of critical clinical variables such as diabetes duration and comorbid conditions. Additionally, stratification based on sex and ethnicity is essential. Furthermore, exploring additional polymorphisms within the *ACE* gene and associated pathways could yield a more nuanced understanding of the genetic underpinnings of diabetic neuropathy.

## 5. Limitations and Future Directions

This review has some limitations that warrant consideration. Firstly, the relatively small number of eligible studies identified (n = 11) limits the generalizability of our conclusions. A significant number of initially screened studies were excluded due to inadequate methodological rigor, insufficient genotypic detail, or unavailability of complete texts. Furthermore, most of the included studies were conducted in geographically restricted regions, primarily in South Asia and the Middle East, with minimal representation from Latin America, Africa, and other populations. This geographic concentration diminishes the ethnic and genetic diversity in the analyzed cohorts, which may distort the interpretation of allele and genotype distribution patterns and limit the generalizability of findings to the global population. In support of this, a recent systematic review noted that most investigations into DN genetics have primarily focused on White and Arab populations, with only one study including African American individuals, further emphasizing the lack of representation across diverse ethnic groups [16].

Methodological aspects, such as differences in diagnostic criteria for DN, sample sizes, and genotyping techniques, may partly explain these discrepancies. The evaluation of genotyping methodologies revealed that, although all studies focused on the same polymorphism, the methods employed were diverse, including conventional PCR [18,38,39], Taq-PCR [17,30,31,32], PCR-RFLP [33,35], and PCR-MADGE [29]. While all investigations centered on the *ACE* I/D polymorphism, the diversity in genotyping methodologies, diagnostic criteria for DN, and patient stratification approaches varied significantly. Such discrepancies complicate direct comparisons and may help explain the divergent results observed across different cohorts. Furthermore, many studies failed to adequately adjust for confounding variables, including diabetes duration, glycemic control, and hypertension. Despite the validation of these methods, variations in methodology could lead to discrepancies in the consistency and reproducibility of results across different studies [36,37,42].

Another significant source of heterogeneity stems from the diagnostic criteria for DN. Various investigations employed validated symptom assessment tools or adhered to international consensus criteria. For example, Ramanathan utilized the Michigan Neuropathy Screening Instrument (MNSI) in conjunction with clinical and neurophysiological assessments [20]. At the same time, Inanir implemented the Neuropathy Symptom Score (NSS) and the Neuropathy Disability Score (NDS) [34]. Stephens adhered to established international consensus definitions (Boulton/Gries/Jervell; ADA San Antonio), necessitating the identification of symptoms or signs corroborated by at least two objective findings, such as monofilament, pinprick, vibration, joint position sense, or ankle reflexes, following the exclusion of alternate etiologies [31]. In a similar vein, Ito diagnosed diabetic polyneuropathy using typical bilateral sensory symptoms paired with vibration perception threshold assessments via tuning fork, designating sensory times exceeding five seconds as abnormal [29].

Some studies included neurophysiological validation and combining clinical evaluations with electrophysiological assessments. Jurado’s approach involved a clinical neurological examination supplemented with nerve conduction studies (NCS) in cases requiring further confirmation, analyzing the median and ulnar nerves for the upper extremities, as well as the sural, peroneal, and tibial nerves for the lower extremities [32]. Degirmenci adopted a composite strategy that included comprehensive neurological examinations, autonomic function tests such as deep breathing and the 30:15 ratio, along with electrophysiological measurements, to define neuropathy based on sensory, motor, or autonomic involvement [30]. Amiri followed a similar methodology, integrating clinical and neurophysiological evaluations [37]. A smaller number of studies specifically addressed CAN. Abdalrada leveraged data from the Australian DiScRi project, which encompassed a wide array of diagnostic protocols, including the Ewing battery, heart rate variability assessments, ECG monitoring, retinal evaluations, and peripheral nerve function analyses, alongside demographic and biochemical data [36].

In contrast, Dhumad focused on a limited selection of noninvasive tests for CAN diagnosis and acknowledged the consequent limitations, particularly the absence of longitudinal follow-up [19]. However, a few reports provided insufficient detail regarding diagnostic methodologies. Settin reported the prevalence of DN among participants but failed to specify the diagnostic criteria employed [35]. Mansoor indicated that patients with and without DN were screened at a tertiary care facility, but did not clarify the methods or criteria used [33]. Dhumad similarly noted the inadequacy of a comprehensive CAN battery as a limitation [19].

Collectively, these findings underscore the substantial variability in diagnostic criteria for DN, ranging from standardized clinical scoring and consensus definitions to detailed electrophysiological evaluations, as well as instances of vague or omitted methodological reporting. This diversity is a significant source of potential bias and limits the comparability of results across studies. While our scoping review did not incorporate a formal risk-of-bias analysis, it is crucial to recognize that the absence of standardized diagnostic definitions and limited methodological reporting present notable constraints that likely exacerbate the observed heterogeneity in findings. The lack of standardized diagnostic criteria poses a major source of bias and complicates the ability to compare findings across different studies.

An additional important consideration is that while the scientometric analysis provided valuable insights into research trends, it was limited to evaluating keyword co-occurrence and term frequency. The absence of citation network and collaboration analyses represents a missed opportunity to gain a deeper understanding of the knowledge landscape and to identify significant research gaps.

Lastly, an additional significant limitation of this scoping review is that it was intentionally designed without incorporating quantitative synthesis or statistical adjustments. Consequently, we did not conduct normalization across studies, implement formal controls for confounding factors (such as age, sex, and comorbidities), perform sensitivity analyses, harmonize allele frequencies according to specific genetic models, or carry out formal assessments of publication bias. These methodological considerations are crucial, as they can directly impact the robustness and generalizability of the associations observed between ACE polymorphisms and DN. Future systematic reviews that incorporate meta-analyses will be crucial in addressing these deficiencies, facilitating standardized evaluations of genetic models, adjustments for clinical covariates, and formal tests for heterogeneity and publication bias. Such methodologies will complement the descriptive evidence presented here and provide a more robust quantitative foundation for understanding the role of ACE variants in DN.

## 6. Conclusions

This scoping review indicates the existence of a consistent association between the D allele and/or the DD genotype and an increased susceptibility to DN across diverse populations. Conversely, the II genotype appears to function as a protective factor in particular ethnic cohort. However, these associations show considerable variability, underscoring the influence of population-specific genetic backgrounds, comorbidities, environmental factors, and methodological differences. The scientometric analysis corroborates these findings, revealing a limited yet concentrated research landscape that predominantly addresses peripheral neuropathy and *ACE* polymorphisms, while highlighting significant gaps in geographic representation and the diversity of *ACE* genetic variants investigated. Collectively, these insights emphasize the multifaceted nature of DN, where the interplay between genetic susceptibility and clinical variables remains inadequately characterized. Notably, the exclusive focus on the I/D polymorphism across the reviewed studies indicates a clear need for more comprehensive genomic investigations that consider other *ACE* variants and their roles within the renin–angiotensin system and neurovascular homeostasis.

Future research endeavors should emphasize larger, multiethnic cohorts, standardized diagnostic criteria for DN, and the robust control of confounding factors. Integrating genetic data with clinical, biochemical, and environmental variables could significantly enhance risk stratification models and foster the development of precision medicine strategies. Ultimately, deepening our understanding of the genetic architecture underlying DN holds the promise of identifying novel biomarkers and targeted interventions aimed at preventing or alleviating this debilitating diabetes complication.

## Figures and Tables

**Figure 1 diseases-13-00289-f001:**
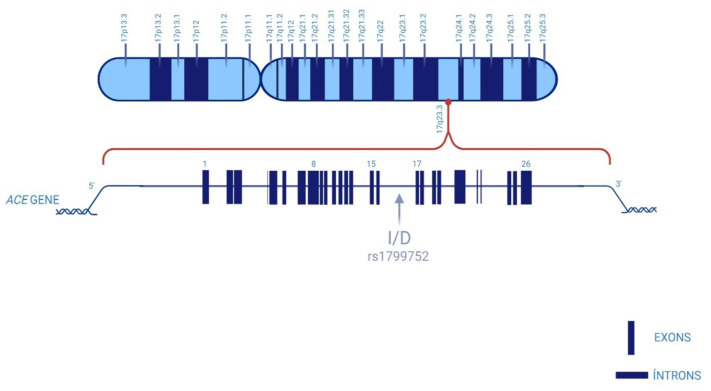
Schematic representation of the insertion/deletion (I/D) polymorphism in intron 16 of the ACE gene, located on the long arm of chromosome 17. The insertion (I) allele contains a 287-base-pair fragment that is absent in the deletion (D) allele, resulting in three possible genotypes: II, ID, and DD. Created with BioRender.com. (24 June 2025).

**Figure 2 diseases-13-00289-f002:**
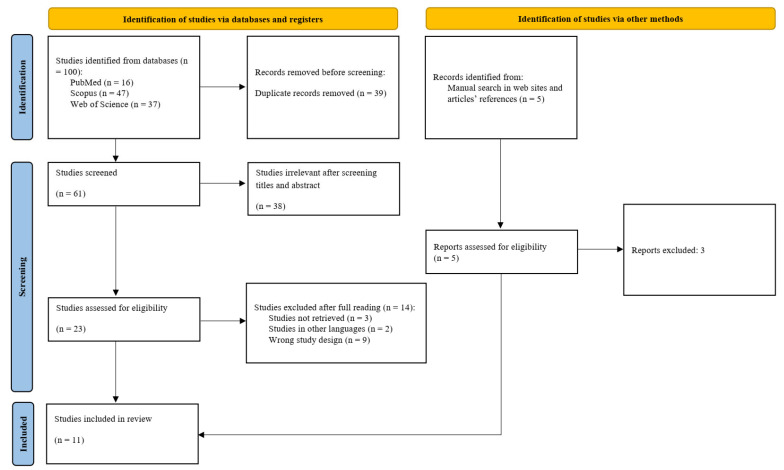
Scoping review flowchart. Source: PRISMA-ScR.

**Figure 3 diseases-13-00289-f003:**
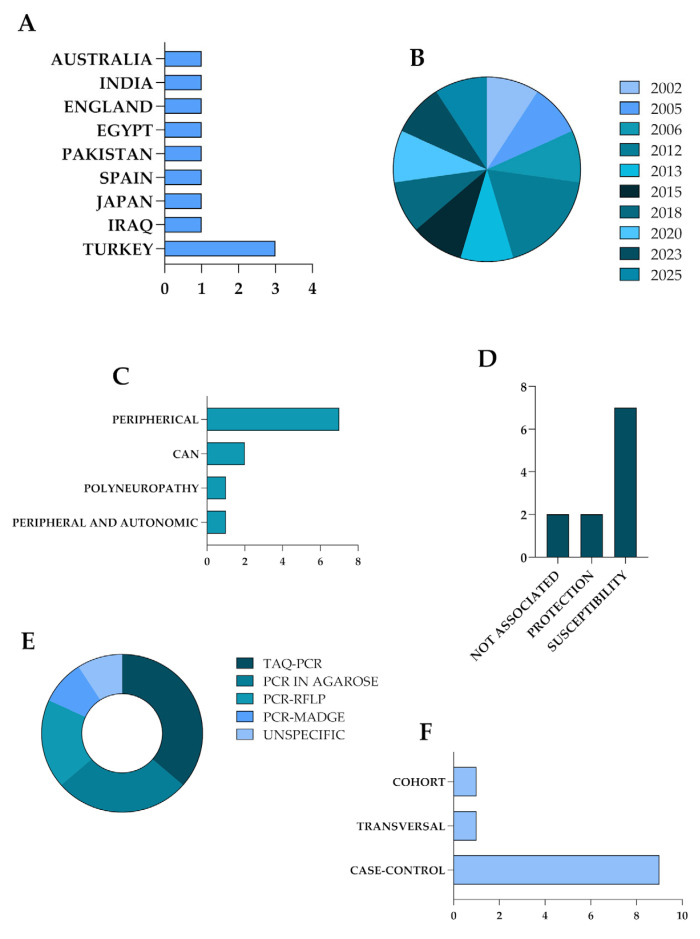
Descriptive overview of the studies included in this review. Panel (**A**) illustrates the geographic distribution of studies by country of origin. Panel (**B**) presents the distribution of publications by year. Panel (**C**) categorizes the type of diabetic neuropathy investigated (peripheral or autonomic). Panel (**D**) demonstrates the distribution of the included studies according to the association identified with the outcome: protection, susceptibility, or no association. Panel (**E**) details the genotyping methodologies employed across the studies. Panel (**F**) illustrates the study designs of the analyzed works. Abbreviations: CAN—cardiac autonomic neuropathy; PCR-RFLP—polymerase chain reaction-restriction fragment length polymorphism; PCR-MADGE—polymerase chain reaction-microplate array diagonal gel electrophoresis.

**Figure 4 diseases-13-00289-f004:**
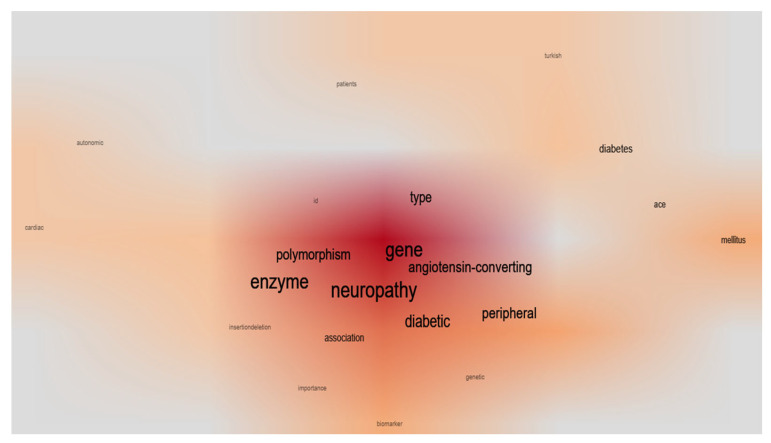
The word cloud visualization illustrates the most frequently cited terms from the included articles, generated through scientometric mapping. The size and color intensity of the font reflect the frequency of term occurrence. This visualization emphasizes the thematic concentration of research on *ACE* gene polymorphisms and DN, highlighting both core areas of focus and underrepresented domains.

**Figure 5 diseases-13-00289-f005:**
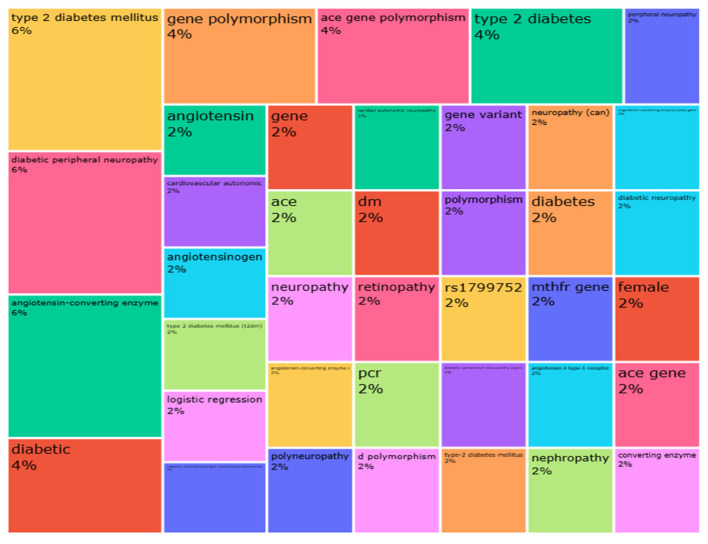
Hierarchical representation of the most relevant terms identified in the included studies. The size of each rectangle reflects the frequency and prominence of the terms across the analyzed literature, highlighting the central themes and focus areas of the selected publications.

**Figure 6 diseases-13-00289-f006:**
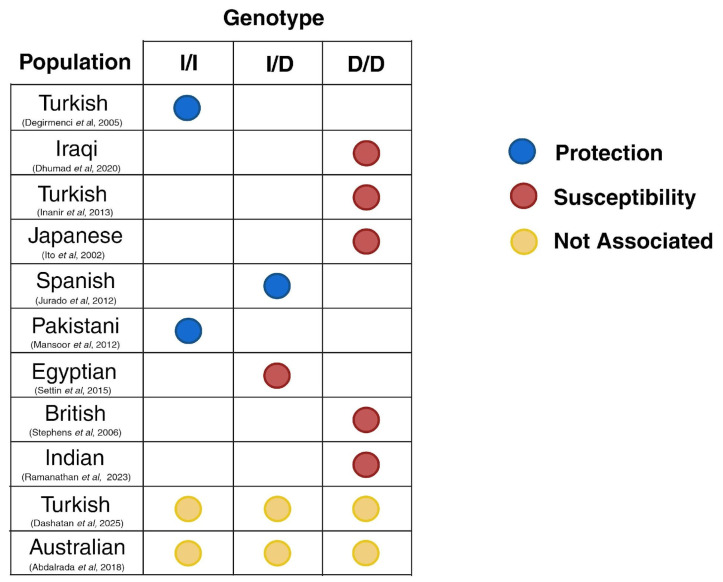
Inter-population variability in the association between ACE I/D genotypes and DN. The schematic depicts qualitative directions reported across studies for each genotype (I/I, I/D, D/D). Blue = protective effect; red = susceptibility; yellow = no significant association. This figure summarizes study-level findings across diverse cohorts, highlighting heterogeneity in reported associations that may stem from differences in genetic background, clinical covariates, and methodological design. This schematic is descriptive only and does not represent pooled effect sizes [19,20,29,30,31,32,33,34,35,36,37].

**Table 1 diseases-13-00289-t001:** Research strategies for PubMed, Scopus and Web of Science.

Database	Queries
PubMed	((((((Polymorphism, Single Nucleotide[MeSH Terms]) OR (Polymorphism*[Title/Abstract])) OR (“SINGLE NUCLEOTIDE POLYMORPHISM”[Title/Abstract])) OR (SNP[Title/Abstract])) AND Diabetes Mellitus”[MeSH Terms]) OR (“Diabetes complications”[MeSH Terms])) OR (“DIABETES MELLITUS”[Title/Abstract])) OR (DM[Title/Abstract])) OR (Diabetic[Title/Abstract])) OR (“Diabetic complications”[Title/Abstract]))) AND ((((((((((((neuropathy[Title/Abstract]) OR (Painful[Title/Abstract])) OR (“peripheral neuropathy”[Title/Abstract])) OR (polyneuropathy[Title/Abstract])) OR (“neuropathic pain”[Title/Abstract]))
Scopus	((TITLE-ABS-KEY (“polymorphism, single nucleotide”) OR TITLE-ABS-KEY (“SINGLE NUCLEOTIDE POLYMORPHISM”) OR TITLE-ABS-KEY (snp) OR TITLE-ABS-KEY (polymorphism*))) AND ((TITLE-ABS-KEY (ace) OR TITLE-ABS-KEY (cd143) OR TITLE-ABS-KEY (ace1) OR TITLE-ABS-KEY (dcp1) OR TITLE-ABS-KEY (dpc) OR TITLE-ABS-KEY (“ANGIOTENSIN CONVERTING ENZYME”) OR TITLE-ABS-KEY (“ace gene*”) OR TITLE-ABS-KEY (“angiotensin-converting enzyme gene”))) AND ((TITLE-ABS-KEY (“diabetic polyneuropath*”) OR TITLE-ABS-KEY (“peripheral neuropathy”) OR TITLE-ABS-KEY (“Diabetic Neuropathy”) OR TITLE-ABS-KEY (“Diabetes-Related Neuropathy”) OR TITLE-ABS-KEY (“Diabetic Peripheral Neuropathy”) OR TITLE-ABS-KEY (“Diabetes-Induced Neuropathy”) OR TITLE-ABS-KEY (“Chronic Diabetic Neuropathy”) OR TITLE-ABS-KEY (“Diabetic Autonomic Neuropathy”) OR TITLE-ABS-KEY (dnp) OR TITLE-ABS-KEY (pdpn) OR TITLE-ABS-KEY (“painful diabetic peripheral neuropathy”)))
Web of Science	“diabetic polyneuropath*” (Topic) or “peripheral neuropathy” (Topic) or “Diabetic Neuropathy” (Topic) or “Diabetes-Related Neuropathy” (Topic) or “Diabetic Peripheral Neuropathy” (Topic) or “Diabetes-Induced Neuropathy” (Topic) or “Chronic Diabetic Neuropathy” (Topic) or “Diabetic Autonomic Neuropathy” (Topic) or DNP (Topic) or PDNP (Topic) or “painful diabetic peripheral neuropathy” (Topic) AND ACE (Topic) or CD143 (Topic) or ACE1 (Topic) or DCP1 (Topic) or DCP* (Topic) or “ANGIOTENSIN CONVERTING ENZYME” (Topic) or ace gene* (Topic) or “angiotensin-converting enzyme gene” (Topic) AND “polymorphism, single nucleotide” (Topic) or “SINGLE NUCLEOTIDE POLYMORPHISM” (Topic) or SNP (Topic) or polymorphism* (Topic)

**Table 2 diseases-13-00289-t002:** General characteristics of the selected studies.

Author/ Year	Study Model	Nationality of Participants	Neuropathy Type	Genotyping Methodology	Sample Size (F/M)	ACE Polymorphism	Association
Ito et al., 2002 [29]	Case–control	Japanese	Polyneuropathy	PCR in agarose gel	84 T2DM (63 without polyneuropathy and 21 with polyneuropathy)	I/D in intron 16 (rs1799752)	The DD genotype was associated with susceptibility to the development of DPN, with the D allele being the susceptibility factor.
Degirmenci et al., 2005 [30]	Case–control	Turkish	Peripheral and autonomic neuropathy	PCR in agarose gel	143 T2DM and 133 controls	I/D in intron 16 (rs1799752)	The ID genotype was associated with an increased risk of developing DN, while the II genotype was related to a protective factor.
Stephens et al., 2006 [31]	Case–control	British	Peripheral neuropathy	PCR-MADGE	1020 recruited; remained 572 (230/342)	I/D in intron 16 (rs1799752)	The D allele is related to microvascular complications in women with T2DM.
Jurado et al., 2012 [32]	Longitudinal cohort	Spanish	Peripheral neuropathy	Taq-PCR in agarose gel	283 patients (108/175) with T2DM, of whom 82 had DPN.	I/D in intron 16 (rs1799752)	The presence of DPN is correlated with age; the presence of the D/I genotype is negatively related to the development of DPN.
Mansoor et al., 2012 [33]	Case–control	Pakistani	Peripheral neuropathy	Taq-PCR in agarose gel	276 T2DM with DPN, 496 T2DM without DPN and 331 controls	I/D in intron 16 (rs1799752)	Genotype II of the ACE gene protects T2DM patients from developing DPN. In comparing T2DM patients with and without DPN, a higher prevalence of women was noted in both groups, indicating a potential link between gender and microvascular complications. Patients with the DD genotype exhibited high serum ACE levels.
Inanir et al., 2013 [34]	Case–control	Turkish	Peripheral neuropathy	Taq-PCR in agarose gel	235 DPN and 281 healthy controls	I/D in intron 16 (rs1799752)	The DD genotype was associated with susceptibility to the development of DPN.
Settin et al., 2015 [35]	Case–control	Egyptian	CAN	PCR-RFLP	202 T2DM (118/84), 93 patients with DPN, and 311 (182/129) controls	I/D in intron 16 (rs1799752)	Polymorphism was associated with a higher susceptibility to T2DM and its microvascular complications.
Abdalrada et al., 2018 [36]	Case–control	Australian	CAN	Not specified	299 patients (132 without CAN and 167 with CAN)	I/D in intron 16 (rs1799752)	There was no significant difference in the distribution of genotypes between patients with or without CAN.
Dhumad et al., 2020 [19]	Cross-sectional	Iraqi	CAN	Taq-PCR in agarose gel	A total of 142 T2DM patients (62/80) and 100 healthy individuals (39/61) were selected. Of the patients with DM, 117 participated in the genetic evaluation, including 62 without CAN and 55 with it, along with 75 healthy controls.	I/D in intron 16 (rs1799752)	The DD genotype and the D allele in the I/D polymorphism of the *ACE* gene may represent a risk factor for the development of T2DM, and the D allele may be associated with an increased incidence of CAN.
Ramanathan and Velayutham, 2024 [20]	Case–control	Indian	Peripheral neuropathy	PCR in agarose gel	90 (30 T2DM with DNP; 30 T2DM without DPN; 30 controls)	I/D in intron 16 (rs1799752)	The D allele is associated with the risk of developing DPN in T2DM patients.
Dashatan et al., 2025 [37]	Case–control	Turkish	Peripheral neuropathy	PCR-RFLP	140 (90 T2DM [57/33]—50 CTR [25/25]).	I/D in intron 16 (rs1799752)	There was no association.

## Data Availability

The authors confirm that the data supporting the findings of this study are available within the article.

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
