# Peer review of "Angiotensin-Converting Enzyme Gene Polymorphisms and Diabetic Neuropathy: Insights from a Scoping Review and Scientometric Analysis"

_diseases, 2025, doi:10.3390/diseases13090289_

Round 1

Reviewer 1 Report

Comments and Suggestions for Authors

The manuscript is valuable in the context of genetic risk assessment. The authors presented the results and documented the findings of their analyses in a satisfactory manner. References to other polymorphisms, apart from the one described, would be useful in the manuscript. A suggestion for a signalling pathway linking the polymorphism to disease risk, based on the authors' biochemical knowledge, would also be valuable. This type of work also requires a more extensive bibliography, especially more recent references.

Author Response

# REVIEWER 1

The manuscript is valuable in the context of genetic risk assessment. The authors presented the results and documented the findings of their analyses in a satisfactory manner.

Answer: We appreciate the Reviewer’s positive evaluation of our manuscript and acknowledgment of its significance in the realm of genetic risk assessment. We are pleased that both the presentation of our findings and the rigor of our documentation have met the expected standards. We are also grateful for the constructive feedback provided. Your observations contributed significantly to improving the scientific clarity, precision, and robustness of the final version of our work. All suggestions were carefully considered and addressed either directly in the revised manuscript or in the specific responses below. The changes made to the text are clearly highlighted in green in the revised version. They are annotated with indications of the respective page and line numbers for transparency and ease of verification.

Once again, we thank you for your valuable contribution to refining our study.

References to other polymorphisms, apart from the one described, would be useful in the manuscript.

Answer: We appreciate the Reviewer’s insightful suggestion. In response, we have expanded the Introduction with a new paragraph that examines additional polymorphisms associated with diabetic neuropathy (DN), as highlighted in recent literature. These new references enhance the genetic framework and underscore the idea that while various genes may influence DN susceptibility, our review concentrates specifically on the angiotensin-converting enzyme (ACE) I/D polymorphism due to its frequent investigation and clinical significance. The revised text can be found on page 2, lines 74–84 and 89–92  of the updated manuscript, highlighted in green.

A suggestion for a signalling pathway linking the polymorphism to disease risk, based on the authors' biochemical knowledge, would also be valuable.

Answer: We appreciate the insightful suggestion provided. In response, we have expanded the Discussion section to elucidate a mechanistic relationship between the ACE I/D polymorphism and the pathogenesis of DN. Specifically, we detail how the presence of the D allele, particularly the DD genotype associated with elevated ACE activity, enhances the conversion of angiotensinogen to angiotensin II. This increase fosters a milieu of vasoconstriction, oxidative stress, inflammation, and microvascular dysfunction. Collectively, these mechanisms contribute to compromised neural blood flow, thereby heightening the susceptibility of peripheral nerves to injury induced by hyperglycemia. This additional content can be found in the Discussion section on page 14, lines 308–322 of the revised manuscript, highlighted in green.

This type of work also requires a more extensive bibliography, especially more recent references.

Answer: We appreciate the reviewer’s comment and have thoroughly checked each citation, incorporating recent and relevant studies to provide the necessary background for addressing the requested revisions following this initial round of evaluation. We would like to clarify that some references in our manuscript are derived from the studies included in our scoping review, selected based on our predefined eligibility criteria. Our protocol did not impose any temporal restrictions regarding publication year, allowing us to include all eligible studies to ensure a comprehensive mapping of the existing evidence. Additionally, our citation strategy has emphasized seminal and foundational works that provide solid scientific and methodological support for our arguments, discussions, and definitions.

We would like to reiterate that pertinent citations were included to respond to the reviewers' suggestions adequately. We sincerely appreciate the comment and the opportunity to clarify our criteria.

Reviewer 2 Report

Comments and Suggestions for Authors

The article Angiotensin-converting Enzyme gene polymorphisms and dia-2 betic neuropathy: Insights from a Scoping Review and Scien-3 tometric Analysis is very well written article. It follows Preferred Reporting Items for Sys-118 tematic Reviews and Meta-Analyses Extension for Scoping Reviews (PRISMA-ScR) 119 Checklist, as noted in references [25,26].

Still, the only remark is that this article is extremely lengthy, and the majority of text can be presented in more summarized way. 

Here are some comments: 

Abstract 

Line 30-32 – “The scientometric analysis revealed “peripheral diabetic neuropathy,” “type 2 diabetes,” and “ACE gene polymorphism” as the most prominent terms in the literature, highlighting consistent research focus but limited geographic and thematic diversity” 

Comment: Please, try to clarify this sentence 

Introduction 

Line 104-105 – “Nevertheless, the results are inconsistent across different populations, likely arising from ethnic variability, methodological differences, and clinical heterogeneity.” 

Comment: is ethnic variability or genetic diversity?

Discussion

Line 247 – please replace term “covered” with included.  

Line 275 – please replace term "expression the DD” with DD. 

Line 333 – please specify the term “notable correlation” with quantitative values. 

Figure 6 – consider excluding Figure 6 from Discussion and consider including this information in Discussion section in narrative way. 

Line 410 – is geographic bias genetic diversity, ethnic background or sampling bias ?   Please specify 

Conclusion  

Line 430 -431 – This sentence “This scoping review systematically maps and critically evaluates the current evidence regarding the association between ACE gene polymorphisms and DN, with a specific focus on the insertion/deletion (I/D) variant located in intron 16” is repetitive and not necessary in Conclusion. 

Author Response

# REVIEWER 2

The article Angiotensin-converting Enzyme gene polymorphisms and dia-2 betic neuropathy: Insights from a Scoping Review and Scien-3 tometric Analysis is very well written article. It follows Preferred Reporting Items for Sys-118 tematic Reviews and Meta-Analyses Extension for Scoping Reviews (PRISMA-ScR) 119 Checklist, as noted in references [25,26].

Still, the only remark is that this article is extremely lengthy, and the majority of text can be presented in more summarized way. 

Answer: We appreciate the Reviewer’s constructive feedback. In response, we have made careful revisions to the manuscript aimed at reducing redundancies and enhancing the clarity of our findings. We have condensed the sections on Results, Discussion, and Conclusion to improve readability and eliminate unnecessary repetition. However, we retained the integrity of the Introduction and Methods sections, as they provide essential context for understanding the rationale, guiding research questions, objectives, and the methodology employed.

Additionally, we would like to highlight that, beyond this remark, other reviewers suggested further expansions, such as the inclusion of references to additional polymorphisms and mechanistic pathways. We endeavored to address all these valuable suggestions while being mindful not to make excessive cuts that could undermine coherence or scientific rigor. Consequently, the revised manuscript strikes a balance between conciseness and completeness, ensuring clarity for the reader while duly incorporating all reviewers' insights. We trust that these adjustments have improved the balance between clarity and detail in the manuscript. All modifications are highlighted in green in the revised version.

Here are some comments: 

Abstract 

Line 30-32 – “The scientometric analysis revealed “peripheral diabetic neuropathy,” “type 2 diabetes,” and “ACE gene polymorphism” as the most prominent terms in the literature, highlighting consistent research focus but limited geographic and thematic diversity” 

Comment: Please, try to clarify this sentence 

Answer: We thank the Reviewer for pointing out this lack of clarity. The sentence was revised for greater precision and readability. The new version now reads:

“The scientometric analysis identified ‘peripheral diabetic neuropathy,’ ‘type 2 diabetes,’ and ‘ACE gene polymorphism’ as the most frequently co-occurring terms, indicating that research on this topic has been concentrated around these themes, while showing limited diversity in geographic origin and scope.”

This revised phrasing appears highlighted in green (Abstract, page 1).

Introduction 

Line 104-105 – “Nevertheless, the results are inconsistent across different populations, likely arising from ethnic variability, methodological differences, and clinical heterogeneity.” 

Comment: is ethnic variability or genetic diversity?

Answer: We thank the Reviewer for this relevant remark. The original wording “ethnic variability” was revised to “genetic diversity among populations,” which more accurately reflects the underlying factor contributing to heterogeneous results in association studies. The revised sentence now reads:

“Nevertheless, the results are inconsistent across different populations, likely arising from genetic diversity, methodological differences, and clinical heterogeneity.”

This adjustment was made in the Introduction, on page 3, lines 120–123 of the revised manuscript, and is highlighted in green.

Discussion

Line 247 – please replace term “covered” with included.  

Answer: We thank the Reviewer for this suggestion. The modification was performed and highlighted in green in the revised manuscript.

Line 275 – please replace term "expression the DD” with DD. 

Answer: We thank the Reviewer for this suggestion. The modification was performed and highlighted in green in the revised manuscript.

Line 333 – please specify the term “notable correlation” with quantitative values. 

Answer: We thank the Reviewer for this observation. We agree that the wording “notable correlation” could inadvertently suggest a quantitative statistical analysis, which was not performed in this scoping review. To avoid misinterpretation, we have revised the sentence to read:

“Collectively, the analysis of these studies indicates a link between the DD genotype and an elevated risk of developing DN, suggesting that the D allele may act as a susceptibility factor.”

This adjustment clarifies our intention to emphasize a qualitative pattern observed across studies rather than a formal correlation analysis. The revised text can be found in the Discussion, on page 16, lines 393-395 of the manuscript, highlighted in green.

Figure 6 – consider excluding Figure 6 from Discussion and consider including this information in Discussion section in narrative way. 

Answer: We appreciate the Reviewer’s suggestion. We would like to respectfully propose the retention of Figure 6, as it offers a concise visual synthesis of genotype–phenotype relationships across populations, a complexity that is challenging to articulate with the same clarity in narrative form. This figure complements the text by facilitating a quick cross-study comparison of I/I, I/D, and D/D genotypes in relation to DN direction (protection/susceptibility/no association).

To address the concern and eliminate any ambiguity, we have made the following enhancements:

  1. Expanded the figure's caption to ensure it is fully self-contained, clearly stating that this is a qualitative mapping (with no pooled effect sizes implied).
  2. Included an explicit cross-reference in the Discussion to direct readers to the narrative that interprets these observed patterns.

The revisions are highlighted in green in the updated document (Discussion section, Page 16, lines 407-416).

Line 410 – is geographic bias genetic diversity, ethnic background or sampling bias ?   Please specify 

Answer: We appreciate the Reviewer for highlighting this important concern. We recognize that the original term "geographic bias" may have been unclear. To clarify, we intended to indicate the limited geographic origins of the included studies, which were predominantly conducted in South Asia and the Middle East. This limitation results in the underrepresentation of populations from other regions, thereby limiting the diversity of ethnic and genetic backgrounds in the existing evidence. To prevent any misinterpretation, we have revised the text and included a brief statement about this topic. This revision can be found in the Limitations and Future Directions section, page 18, lines 481–490 (highlighted in green).

Conclusion  

Line 430 -431 – This sentence “This scoping review systematically maps and critically evaluates the current evidence regarding the association between ACE gene polymorphisms and DN, with a specific focus on the insertion/deletion (I/D) variant located in intron 16” is repetitive and not necessary in Conclusion. 

Answer: We appreciate your comment. The sentence was removed.

Reviewer 3 Report

Comments and Suggestions for Authors

The study aimed to synthesize the scientific evidence on ACE gene polymorphisms and their association with DN through a scoping review combined with scientometric analysis. Eleven observational studies were included, with the D allele and DD genotype associated with increased DN susceptibility in over half of the reports, while the II genotype appeared protective in some cases. The work addresses a clinically relevant question and combines bibliometric mapping with genetic association analysis. However, several points require clarification to strengthen the manuscript:

Can the authors provide more detailed information on the specific diagnostic criteria and classification used for selecting patient cohorts?

Could the authors clarify the normalization methods applied in the data analysis, particularly for inter-sample variability?

How were potential confounding factors such as age, sex, or comorbidities addressed in the statistical analysis?

Were any sensitivity analyses performed to confirm the robustness of the results?

Were the allele frequencies from different studies harmonized using the same genetic model (dominant, recessive, or additive) before comparison?

Was any assessment (e.g., funnel plot, Egger’s test) performed to check for publication bias in the included literature?

Regarding Figure 4: the font size and color intensity represent the most frequently cited terms in the included articles. Please clarify whether this visualization is essential for the main conclusions or if it could be moved to supplementary material.

Author Response

#REVIEWER 3

The study aimed to synthesize the scientific evidence on ACE gene polymorphisms and their association with DN through a scoping review combined with scientometric analysis. Eleven observational studies were included, with the D allele and DD genotype associated with increased DN susceptibility in over half of the reports, while the II genotype appeared protective in some cases. The work addresses a clinically relevant question and combines bibliometric mapping with genetic association analysis. However, several points require clarification to strengthen the manuscript:

Answer: We sincerely thank the Reviewer for the favorable evaluation of our study. We also appreciate the valuable suggestion that several points need further clarification. Below, we provide detailed, point-by-point responses and have revised the manuscript accordingly to improve clarity and rigor. All modifications are highlighted in green in the revised version.

Can the authors provide more detailed information on the specific diagnostic criteria and classification used for selecting patient cohorts?

Answer: We appreciate the Reviewer for highlighting this request. In response, we have incorporated a paragraph into the section “Limitations and Future Directions” that summarizes and categorizes the included studies based on their diagnostic approaches for DN. This addition aims to clarify the heterogeneity of definitions and methodologies across studies and how these variations may impact comparability. All modifications are highlighted in green in the revised version (Pages 18, lines 504-514, and 19, lines 515-548).

Could the authors clarify the normalization methods applied in the data analysis, particularly for inter-sample variability?

Answer: We appreciate the Reviewer highlighting this point. We would like to clarify that our study was designed as a scoping review coupled with a scientometric analysis; consequently, no quantitative normalization of biological or clinical samples was performed. The examination of inter-study variability was qualitative and descriptive, emphasizing how individual studies defined diabetic neuropathy, the genotyping methods utilized, and the direction of associations reported for the ACE I/D polymorphism.

In the scientometric component, we employed bibliometric mapping tools (specifically, the Bibliometrix R package) that generate co-occurrence and thematic maps based on publication metadata, including keywords, authorship, and country affiliations. These analyses adhere to standardized procedures within the software and do not entail normalization of patient-level or laboratory data.

To prevent any potential misinterpretation, we have clarified this aspect in the Methods section (pages 6, lines 195-205) and in the Limitations and Future Directions section (page 19, lines 554-567). All modifications are highlighted in green in the revised version.

Lastly, we acknowledge the importance of conducting more robust analyses in the future. We plan to develop systematic reviews accompanied by meta-analyses, which will enable us to test the magnitude of associations formally and statistically account for inter-study variability. Such approaches will serve to provide more compelling quantitative evidence regarding the influence of ACE polymorphisms on diabetic neuropathy.

How were potential confounding factors such as age, sex, or comorbidities addressed in the statistical analysis?

Answer: We thank the Reviewer for raising this important question. Given that our research was conducted as a scoping review accompanied by a scientometric analysis, we did not perform pooled statistical analyses or adjust for confounding factors, such as age, sex, or comorbidities. Instead, we systematically extracted and reported whether the individual studies considered these variables in their design or analysis. As detailed in the Results section and discussed thoroughly in the Discussion, the treatment of confounders varied significantly across studies. While some research stratified by sex or adjusted for age and comorbidities, many others lacked comprehensive information or utilized standardized methodologies.

This inconsistency in controlling for potential confounding factors presents a significant limitation in the existing literature, as variations in clinical covariates (such as duration of diabetes, glycemic control, and hypertension) can substantially impact both neuropathy risk and the expression of ACE genotypes. We have addressed this limitation in the section concerning Limitations and Future Directions. To prevent any potential misinterpretation, we have clarified this aspect in the Methods section (pages 6, lines 195-205) and in the Limitations and Future Directions section (page 19, lines 554-567). All modifications are highlighted in green in the revised version.

Lastly, as noted in our response to the previous comment, future systematic reviews and meta-analyses will be essential for formally assessing the influence of confounders and providing adjusted estimates of association.

Were any sensitivity analyses performed to confirm the robustness of the results?

Answer: We appreciate the Reviewer for this observation. We would like to clarify that, as this is a scoping review combined with a scientometric analysis, we did not perform sensitivity analyses in the statistical sense, as no quantitative synthesis or pooled effect estimates were generated. Instead, our method for assessing the robustness of the evidence was qualitative: we compared findings across different populations, genotyping methods, and other relevant aspects, highlighting both consistencies (such as the recurrent associations of the D allele/DD genotype with diabetic neuropathy in several cohorts) and divergences (including null or protective associations observed in other groups).

To prevent any potential misinterpretation, we have clarified this aspect in the Methods section (pages 6, lines 195-205) and in the Limitations and Future Directions section (page 19, lines 554-567). All modifications are highlighted in green in the revised version.

Were the allele frequencies from different studies harmonized using the same genetic model (dominant, recessive, or additive) before comparison?

Answer: We appreciate the Reviewer for this inquiry. Our study was designed as a scoping review combined with a scientometric analysis; therefore, we did not harmonize allele frequencies across the various studies or re-analyze the data under a single genetic model (such as dominant, recessive, or additive). Instead, we extracted and reported the associations as presented by the original authors, whose choices of genetic models and analytic approaches varied. This qualitative mapping enabled us to highlight instances where studies consistently reported signals for the D allele or DD genotype, as well as cases where null or protective effects were observed, all without altering or reinterpreting the original statistical frameworks.

We also recognize that this heterogeneity in genetic models is one of the factors contributing to the variability of results. In future work, particularly systematic reviews with meta-analysis, we intend to formally standardize allele frequencies and compare genetic models to produce pooled estimates. To prevent any potential misinterpretation, we have clarified this aspect in the Methods section (pages 6, lines 195-205) and in the Limitations and Future Directions section (page 19, lines 554-567). All modifications are highlighted in green in the revised version.

Was any assessment (e.g., funnel plot, Egger’s test) performed to check for publication bias in the included literature?

Answer: We appreciate the Reviewer for highlighting this observation. As this work was conducted as a scoping review combined with a scientometric analysis, we did not undertake a formal assessment of publication bias (e.g., funnel plots, Egger’s test). Such approaches necessitate a quantitative synthesis of effect sizes, which was not applicable in our study, as we did not conduct a meta-analysis. Nevertheless, we qualitatively addressed the potential for publication bias. Our search strategy was designed to be comprehensive, encompassing three major databases (PubMed, Scopus, Web of Science) without any restrictions on publication year. Additionally, the scientometric analysis provided a visualization of the geographic and thematic distribution of the published literature, revealing specific imbalances, such as the predominance of studies from South Asia and the Middle East and the underrepresentation of populations from Latin America and Africa. These patterns may indirectly suggest publication bias or a concentration of research. To prevent any potential misinterpretation, we have clarified this aspect in the Methods section (pages 6, lines 195-205) and in the Limitations and Future Directions section (page 19, lines 554-567). All modifications are highlighted in green in the revised version.

Regarding Figure 4: the font size and color intensity represent the most frequently cited terms in the included articles. Please clarify whether this visualization is essential for the main conclusions or if it could be moved to supplementary material.

Answer: We would like to express our gratitude to the reviewer for their detailed evaluation of our manuscript and for recognizing the clinical importance of our subject matter. In response to the feedback provided, we have conducted a thorough revision of the manuscript to address the identified concerns, enhancing both clarity and structure.

All comments have been meticulously addressed, as outlined in the accompanying letter.

We believe that the revised version effectively addresses the feedback received, providing a more transparent and more rigorous presentation of our findings. All changes have been clearly marked within the manuscript and are referenced by page and line numbers for ease of review.
